# Research on Energetic Micro-Self-Destruction Devices with Fast Responses

**DOI:** 10.3390/mi14050961

**Published:** 2023-04-28

**Authors:** Wenxing Kan, Jie Ren, Hengzhen Feng, Wenzhong Lou, Mingyu Li, Qingxuan Zeng, Sining Lv, Wenting Su

**Affiliations:** 1The School of Mechatronical Engineering, Beijing Institute of Technology, Beijing 100081, China; kanwenxing@bit.edu.cn (W.K.); fenghz@bit.edu.cn (H.F.); zengqingxuan@bit.edu.cn (Q.Z.);; 2Science and Technology on Electromechanical Dynamic Control Laboratory, Beijing Institute of Technology, Beijing 100081, China

**Keywords:** information security, micro-self-destruction device, energetic material, Ni-Cr bridge initiator

## Abstract

Information self-destruction devices represent the last protective net available to realize information security. The self-destruction device proposed here can generate GPa-level detonation waves through the explosion of energetic materials and these waves can cause irreversible damage to information storage chips. A self-destruction model consisting of three types of nichrome (Ni-Cr) bridge initiators with copper azide explosive elements was first established. The output energy of the self-destruction device and the electrical explosion delay time were obtained using an electrical explosion test system. The relationships between the different copper azide dosages and the assembly gap between the explosive and the target chip with the detonation wave pressure were obtained using LS-DYNA software. The detonation wave pressure can reach 3.4 GPa when the dosage is 0.4 mg and the assembly gap is 0.1 mm, and this pressure can cause damage to the target chip. The response time of the energetic micro self-destruction device was subsequently measured to be 23.65 μs using an optical probe. In summary, the micro-self-destruction device proposed in this paper offers advantages that include low structural size, fast self-destruction response times, and high energy-conversion ability, and it has strong application prospects in the information security protection field.

## 1. Introduction

With the ongoing developments in information technology, information security is receiving increasing attention. To improve the security of core information and prevent leakage of this content, many institutions have proposed higher security requirements for the use of core chips [1,2,3,4]. The common security protection measures used in these chips are information encryption, module demagnetization, and built-in anti-leakage programs [5,6]. However, with the continuing development of highly technological methods, such as password decryption technology, the protection measures described above cannot meet the chip security requirements. Therefore, the physical destruction of chips is now considered to be a more efficient and safer anti-information leakage method. The Defense Advanced Research Projects Agency of the US Department of Defense launched the Disappearance of Programmable Resource project [7], which was dedicated to research into the self-destruction control of electronic devices. The purpose of this project was to prevent advanced technology leakage and loss of military advantage caused by reverse engineering and imitation [8]. In 2012, Hwang et al. designed a biodegradable complementary metal-oxide-semiconductor (CMOS) chip. This CMOS chip was coated with silk protein. When the silk protein was dissolved, the internal soluble metal content of the chip would then dissolve within 2 h, causing the chip to fail [9]. In 2014, Banerjee used a low-temperature post-treatment and micro-packaging process to cut out a concave shape on the back of an off-the-shelf chip and filled in the groove with an expansion material. Heating the expansion material caused it to expand rapidly and produce the stress that destroyed the chip [10]. In 2015, Xerox PARC constructed a self-destructing chip based on glass. Triggered by either a laser or a wireless signal, the resistance element inside the chip would generate Joule heat that broke the glass substrate and circuit to cause complete self-destruction [11]. In 2017, Yoon fabricated a flexible energetic self-destructing chip composed of nitrocellulose paper, a carbon nanotube transistor network, and a silver resistance heater. When this chip was used without authorization, the resistance heater would generate Joule heat that ignited the nitrocellulose paper and destroyed the chip completely [12]. In 2021, the City University of Hong Kong reported new progress in the development of transient microchips [13]. An explosive nano-energetic film was prepared via the in situ growth of nanoscale energetic coordination polymers on graphene oxide sheets. The film was then excited using a resistive microheater as a heat source. A chip containing this film can self-destruct completely within 1 s. The United States and Europe use nano-thermites, such as Al/CuO nano-mixtures, to integrate with silicon chips to form on-chip energetic self-destruct chips. The self-destruct time of such self-destruct chips ranges from milliseconds [14], seconds [15] to tens of seconds [16], and nanothermites need to be integrated inside the chip before packaging [17]. However, this response time for self-destruction is not fast enough and still risks information leakage. Therefore, the need to shorten the self-destruction response time of such chips is an important indicator of information security issues. In 2022, the Beijing Institute of Technology proposed a chip self-destruction module that was composed of a semiconductor bridge (SCB) initiator and a copper azide element [18,19,20]. When a pulse voltage was applied to the SCB, the bridge areas generated high-temperature plasma that detonated the copper azide, generating a detonation wave on the scale of several GPa and causing the chip to be completely destroyed. The driven conditions and time scale of different self-destruction devices are shown in Table 1. When compared to other types of self-destruction methods, the use of detonation waves from energetic materials can cause irreversible damage to the target chip while offering obvious advantages in terms of response time.

Furthermore, with the ongoing innovation in micro-electromechanical systems (MEMS) technology, development is proceeding toward the miniaturization and integration of both the initiator and the energetic charge [21,22], and these new forms have a stable performance and adjustable output energy characteristics. In the energetic micro self-destruction chip, the main functional components are the microstructure initiator and the micro-charge. The microstructure initiator converts input electrical energy into heat energy, and the micro-charge is detonated by the initiator to generate a detonation wave that causes irreversible damage to the chip. The response time of the micro-initiator determines the self-destruction response time directly, and its output energy determines the reliability of the charge detonation process. At present, the research into micro-self-destruction devices driven by the detonation energy of energetic materials is mainly focused on how to improve the output energy and enhance the safety control of these devices, while there have been few studies of energy matching and the response times of energetic micro-self-destruction devices. In this paper, Ni-Cr microstructure initiators with different bridge types and bridge resistance values are tested and compared using an electrical explosion test system. The output energy of the copper azide explosive is determined via simulation calculations and the response time testing of the micro self-destruction device is completed.

## 2. Model and Method

### 2.1. Design of Self-Destruction System

This paper proposes an energetic micro-self-destruction system for information security protection. The proposed system includes an energetic self-destruction module and an energy storage control circuit. The energetic self-destruction module includes a Ni-Cr bridge (NCB) initiator, copper azide, and an external package shell, and this module carries the damage function for the core information chip. The energy storage control circuit includes an energy storage capacitor and other electrical components that are responsible for energy storage and provides the power supply for the self-destruction module. A schematic diagram of the energetic micro-self-destruction system is shown in Figure 1.

When the self-destruction program is to be executed, the energy storage capacitor discharges after storing energy under the action of the driving power supply. An instantaneous high pulse current passes through the NCB’s bridge area, causing the bridge area to complete a physical transformation from the solid state into the plasma state within a µs-scale time. The electrical explosion reaction occurs, the high-temperature plasma is released, and the copper azide is then detonated. The GPa-level detonation wave generated in this process can cause irreversible damage to the information storage chip and thus ultimately ensure information security.

### 2.2. Preparation of the NCB

The metal film bridge initiator uses Joule heat generated by electrical energy to melt, vaporize, and generate plasma within the bridge area, and the electrical explosion then occurs. The working principle of the energetic micro-self-destruction chip is to use the metal film bridge to generate high-temperature plasma under the action of the instantaneous high current from the capacitor discharge, and then detonate the copper azide to generate a detonation wave.

Among the bridge components, the Ni-Cr thin film has characteristics that include high resistivity, a low-temperature coefficient of resistance, good mechanical properties, good thermal stability, and easily drawn materials, and the method used for Ni-Cr thin film preparation is relatively mature [23,24]. This structure is widely used for thin metal film bridge microstructure initiators.

In this study, the NCB was prepared via a magnetron sputtering process. The composition of the Ni-Cr alloy target was 80% Ni and 20% Cr. The NCB structure was then fabricated using an etching process. The preparation flow chart for the NCB is shown in Figure 2.

### 2.3. Electrical Explosion Test System

When the power supply has charged the capacitor completely, the initiation circuit switch is closed to enable conduction through the entire initiation circuit. The high-pulse current passes through the bridge region of the NCB, which causes the bridge region to complete the transition of its physical form from the solid state to the plasma state in a very short time, and the electrical explosion reaction then occurs.

The electro-explosive performance of the NCB at a charging voltage of 20 V was tested using a 100 μF tantalum capacitor as an excitation source. The important parameters, including the electrical explosion delay time and the energy released, were measured. During the test, a voltage probe, a current probe, and a digital oscilloscope were used to record the voltage and current changes in the NCB during the electrical explosion process. A schematic diagram of the electrical explosion test device is shown in Figure 3.

### 2.4. Output Capacity Analysis of the Copper Azide Explosive

Copper azide is the energy output element contained in the micro-self-destruction chip. The size of the detonation wave from this element affects the damage effect, which determines whether the target chip can be destroyed completely. If the copper azide dosage is too small, the output detonation wave pressure will not be sufficient to destroy the target chip. An excessive dosage will reduce the safety of the service processing of the chip, and the cost will also increase. Therefore, this study uses the LS-DYNA simulation calculation software to simulate the detonation wave output capacity of the copper azide at different doses in the air domain.

In the simulation calculations, the air, the explosion, and the device structure are associated via a fluid-solid coupling algorithm and the arbitrary Lagrange-Euler (ALE) element algorithm. The material model for copper azide is called *MAT_HIGH_EXPLOSIVE_BURN, and the equation of state is *EOS_JWL. The pressure of the explosive detonation product is defined as shown in Equation (1):(1)$$p=A\left(1-\frac{\omega }{R_{1}V}\right)e^{-R_{1}V}+B\left(1-\frac{\omega }{R_{2}V}\right)e^{-R_{2}V}+\frac{\omega E}{V}$$

In Formula (1), *p* is the detonation pressure; *V* is the relative specific volume of detonation products; *E* is the internal energy per unit volume; *A*, *B*, *R*_1_, *R*_2_, and *ω* are material constants. The units for *A*, *B*, and *E* are the same as the pressure unit (Pa), and *R*_1_, *R*_2_, *ω*, and *V* are all dimensionless.

The material model of the air is called *MAT_NULL, and the state equation is called *EOS_LINEAR_POLYNOMIAL. The relationship between the pressure and the volume in the air domain can be described using Equation (2):(2)$$p=C_{0}+C_{1}\mu ++C_{2}\mu^{2}+C_{3}\mu^{3}+\left(C_{4}+C_{5}\mu ++C_{6}\mu^{2}\right)E$$

*C*_0_–*C*_6_ are the coefficients of the state equation of air, *E* is the initial internal energy per unit reference volume, and *μ* is the initial relative volume.

Figure 4 shows the simulation calculation model and the related material parameters [25,26].

With the premise of keeping the diameter (*φ*) of the copper azide element unchanged at 1 mm, the charge height (0.2–3.0 mm) is varied. The detonation wave pressure values for different charge dosages and the detonation wave pressure values in the air domain for different gap sizes are then obtained.

For the case where the assembly gap between the information storage chip and the copper azide element is 0.1 mm, the numerical simulation model and the charge dosage versus pressure curve are shown in Figure 5. When the copper azide dosage is in the 0–0.8 mg range, the pressure increases from 1.41 GPa to 1.78 GPa, and the pressure value thus increases greatly. When the copper azide dosage is increased to 1.3 mg, the pressure only increases from 1.78 GPa to 1.82 GPa, and the value thus gradually stabilizes. To ensure that the detonation wave on the chip provides the damage effect required, the pressure value of the detonation wave generated by the copper azide must be greater than the strength of the chip packaging material (1.4 GPa), which means that the copper azide dosage should be more than 0.4 mg.

In addition, the detonation wave pressure values in the air domain for different gap sizes (0.1–0.6 mm) were also analyzed. The pressure curve and the detonation wave transmission process are shown in Figure 6. When the copper azide is detonated, the resulting detonation wave propagates in the vertical direction, and the detonation wave pressure can reach 13 GPa (at 0.07 μs). Subsequently, the detonation wave collides with the constrained shell, and the detonation wave then converges toward the center. The detonation wave pressure then reaches 8.5 GPa (0.28 μs). The converged detonation wave continues to propagate in the vertical direction and the detonation wave pressure in the air domain ranges from 1.1 GPa to 3.4 GPa. The maximum detonation wave pressure appears at a distance of 0.1 mm, and the arrival time of the detonation wave is 0.35 μs. Therefore, setting the assembly gap between the memory chip and the copper azide to be 0.1 mm allows the detonation wave energy of the explosive to be used effectively and improves the reliability of the chip self-destruction process.

## 3. Test and Result

### 3.1. Electric Explosion Performance of the NCB

First, the electrical explosion processes of three NCBs were studied, and the voltage-current (*U*-*I*) curves were obtained. The electrical explosion process of the NCB can be divided into three stages, as shown in Figure 7. This stage from *t*_0_–*t*_1_ represents the film heating, melting, and gasification process. The NCB melts and vaporizes rapidly under the action of the Joule heat, and the voltage reaches a peak at *t*_1_. The bridge temperature rises, and a large amount of gaseous Ni-Cr is ionized to form plasma. When this plasma disappears, the current decreases to zero at *t*_2_, and the electrical explosion reaction is complete.

The instant *t*_1_ is the point at which the vapor generated by vaporization of the metal film is ionized under the action of the voltage, plasma is produced, and the discharge begins. Researchers worldwide define this time as the electrical explosion delay time, and *t*_0_–*t*_1_ represents the electrical explosion delay stage.

The energy generated by the electrical explosion of the NCB is closely related to the ability of the NCB to excite the copper azide. Therefore, it is necessary to calculate the total energy released by the NCB during the electrical explosion process. The calculation formula used is given as follows:(3)$$E_{\mathrm{out}}=\int_{t_{0}}^{t_{2}}U\left(t\right)\cdot I\left(t\right)dt$$

In Equation (3), *E*_out_ is the output energy; *U(t)* is the voltage across the two ends of the NCB; *I(t)* is the current flowing through the NCB; and *t*_0_ and *t*_2_ are the times at which the bridge area begins to power on and power off, respectively.

Figure 8 shows a comparison of the electrical explosion delay time and the total output energy from the three NCBs. The thickness of the three NCBs is the same, and the bridge area increases in turn. Under the same applied charging voltage, the smaller the bridge area is, the less energy required for the phase change is. When the phase change occurs, the required energy can be absorbed in a shorter time, and then the electric explosion reaction occurs faster. However, the smaller the bridge area also means that less energy is released during the electric explosion. Therefore, the electrical explosion delay time and the total energy released by the three NCBs increase with the increase in the bridge area.

Finally, the appearances of the NCB bridge areas after an electrical explosion under charging voltages of 10 V and 20 V are compared. When the charging voltage is too low, the energy stored within the capacitor is insufficient. As a result, the NCB cannot be vaporized completely and cannot thus be converted completely into metal plasma. Residual Ni-Cr material is still present in the bridge area, as shown in Figure 9a, and this residue will ultimately lead to lower total output energy. When the charging voltage increases, the energy stored in the capacitor is sufficient to vaporize the NCB completely and convert it into metal plasma, and the total output energy is also higher, as shown in Figure 9b.

### 3.2. Response Time and Damage Ability Testing of Self-Destruction Device

An optical probe was used to capture the optical signal at the instant at which the copper azide was detonated, and this optical signal was converted into a voltage signal using an oscilloscope. The oscilloscope also collected the voltage signals of the initiator simultaneously. The difference between the rising edge times of these two voltage signals is the response time of the self-destruction device.

In combination with the test results obtained from the electrical explosion testing of the initiator, NCB #1 was selected as the test initiator because it had the shortest electrical explosion delay time and was most conducive to shortening the response time of the self-destruction device. The copper azide dosage in this case was 0.4 mg. The response time and the damage effect test results for the self-destruction device are shown in Figure 10.

In Figure 10, *t*_s_ is the starting discharge time of the initiation circuit, which is the first voltage rise time; *t*_f_ is the optical signal at the instant when the copper azide is detonated, and this is the second voltage rise time. The response time of the self-destructing chip is *t*_f_ minus *t*_s_, which is 23.65 μs. The peak range (*t*_sd_) detected via the optical probe is the action time of the self-destruction device, and the time dispersion range is 5 μs (read value). Therefore, the self-destruction device can damage the target chip within 30 μs. Additionally, the target chip is destroyed completely by the detonation wave from the copper azide, thus achieving the required function of irreversible damage.

## 4. Conclusions

In this paper, detonation waves generated by energetic materials are used to cause irreversible damage to target chips for the purposes of ensuring information security. First, the electrical explosion delay times and the output energies of three Ni-Cr initiators under 20 V and 100 µF capacitor discharges are acquired using an electrical explosion test system. The shortest electric explosion delay time was 16.7 μs and the maximum output energy was 6.2578 mJ. At the same time, the appearances of the bridge areas after the electrical explosion of the Ni-Cr initiator under different charging voltages were compared. When the charging voltage was 10 V, the energy stored in the capacitor was not sufficient to cause the Ni-Cr initiator bridge area to explode completely. Second, the output capacity of the copper azide energetic micro-charge was calculated using LS-DYNA simulation software. When the charge mass is increased, the pressure value of the detonation wave increases gradually in tandem. When the charge mass exceeds 1.3 mg, the change in the pressure value tends to become stable. Following the consideration of the package strength of the target chip, the assembly distance between the copper azide and the chip was set at 0.1 mm, and the detonation wave pressure was 3.8 GPa. Finally, the response time and the damage effect of the energetic micro-self-destruction device were tested. The self-destruction device can cause irreversible damage to the target chip within 30 μs, thereby ensuring information security.

## Figures and Tables

**Figure 1 micromachines-14-00961-f001:**
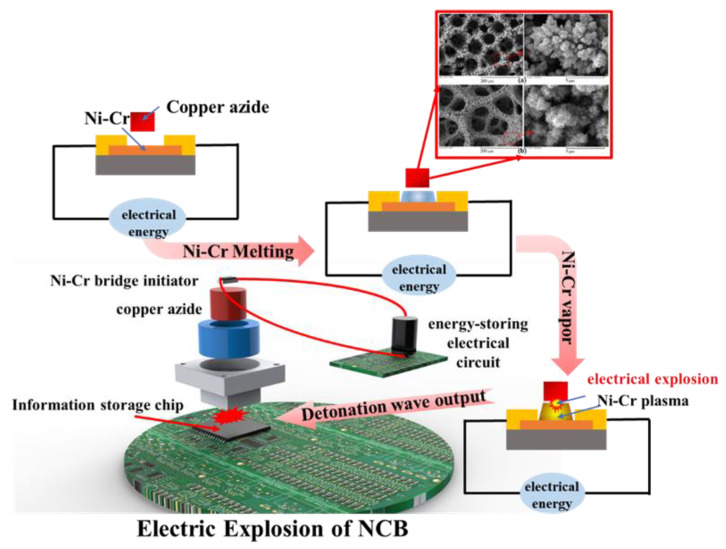
Structural diagram of energetic micro-self-destruction system.

**Figure 2 micromachines-14-00961-f002:**
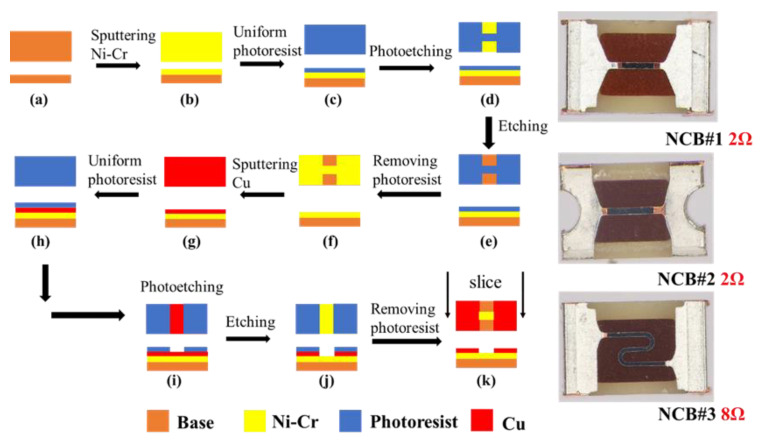
Preparation flow chart for the Ni-Cr bridge (NCB). (**a**) preparing the wafer; (**b**) sputtering Ni-Cr onto a 4-inch glass wafer with a thickness of 900 nm; (**c**) AZ6130 glue with a 3 μm thickness; (**d**) patterning of the bridge area; (**e**) etching of the bridge area; (**f**) cleaning and removing the photoresist; (**g**) sputtering the electrode layer (Cu); (**h**) AZ6130 glue with a thickness of 3 μm; (**i**) patterning of the electrode region; (**j**) etching of the electrode area; (**k**) cleaning and removing the photoresist, scribing, and finally, obtaining the Ni-Cr initiator.

**Figure 3 micromachines-14-00961-f003:**
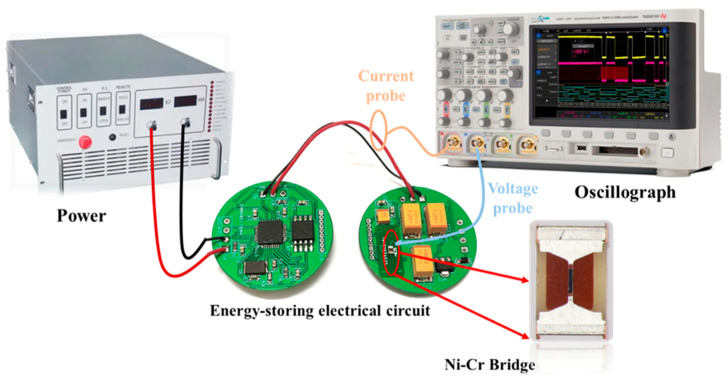
Electrical explosion test system diagram.

**Figure 4 micromachines-14-00961-f004:**
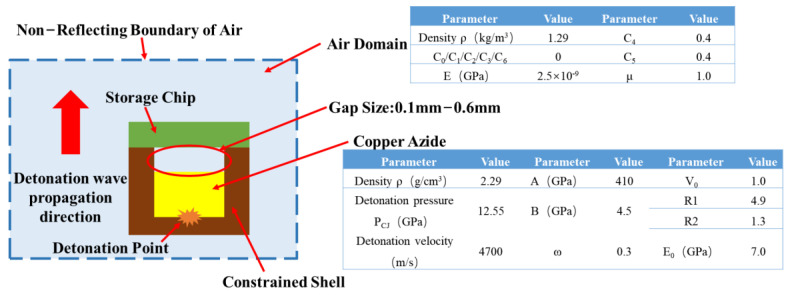
Simulation model and related parameters.

**Figure 5 micromachines-14-00961-f005:**
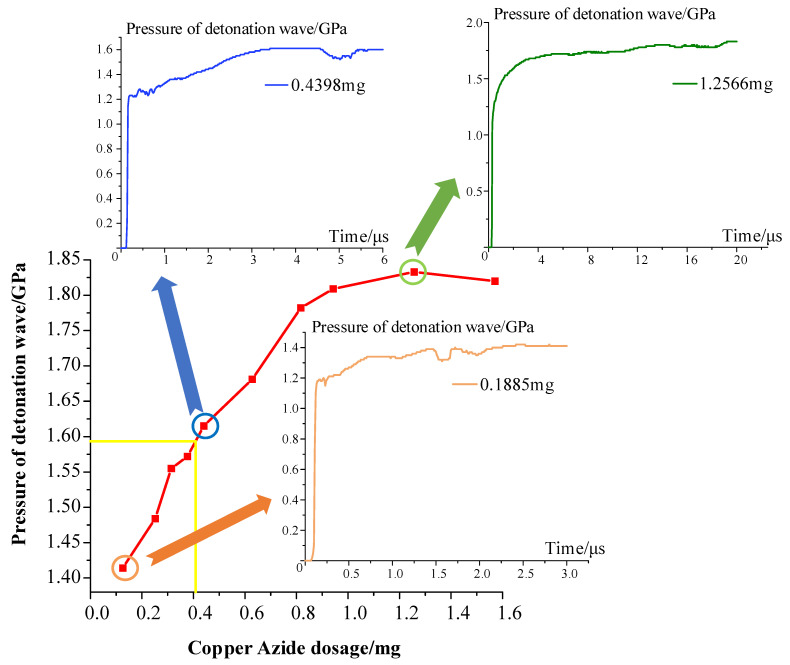
Analysis of copper azide dosage versus detonation wave pressure value.

**Figure 6 micromachines-14-00961-f006:**
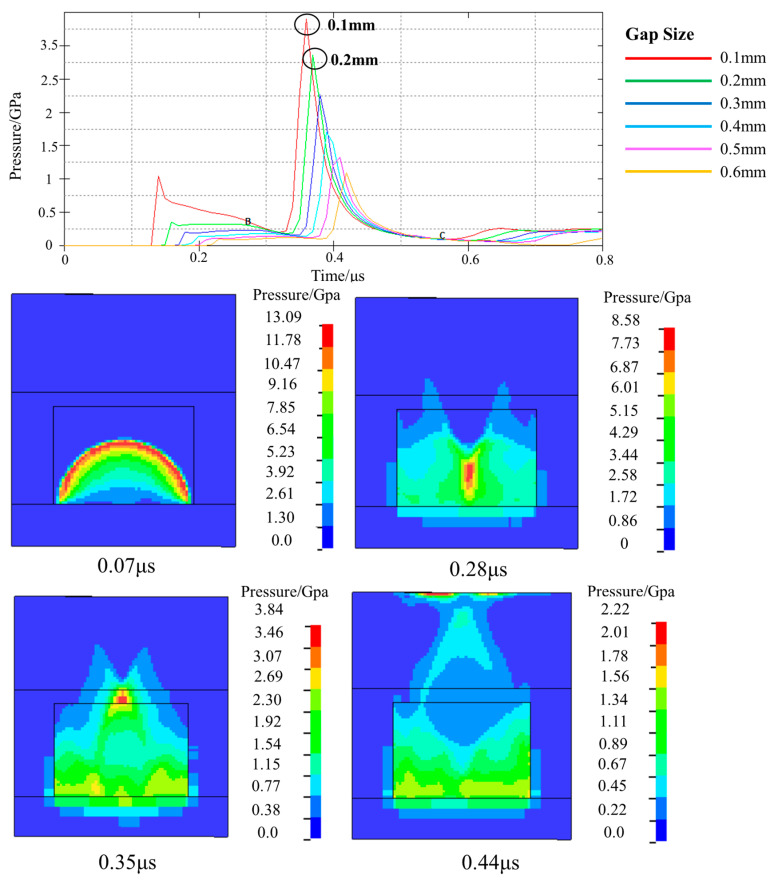
Detonation wave pressure values under different gap conditions and the associated transfer process.

**Figure 7 micromachines-14-00961-f007:**
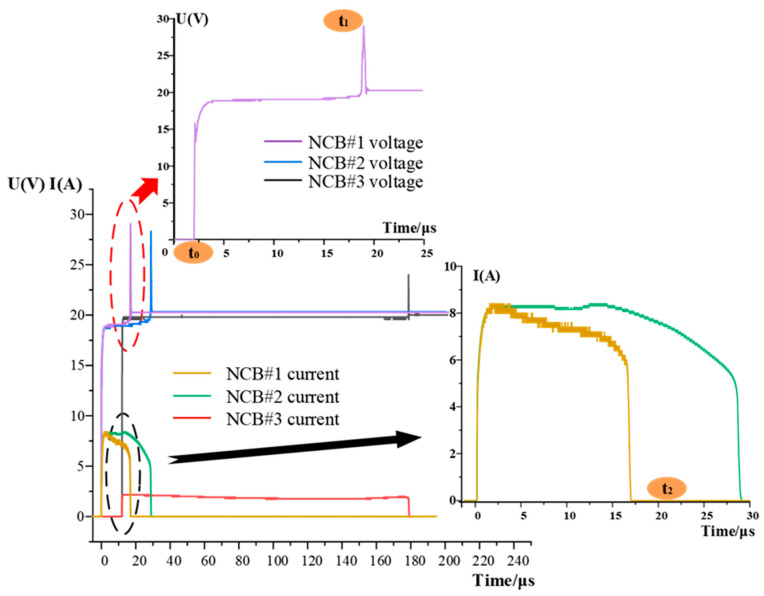
Voltage-current (U-I) curve of the NCB.

**Figure 8 micromachines-14-00961-f008:**
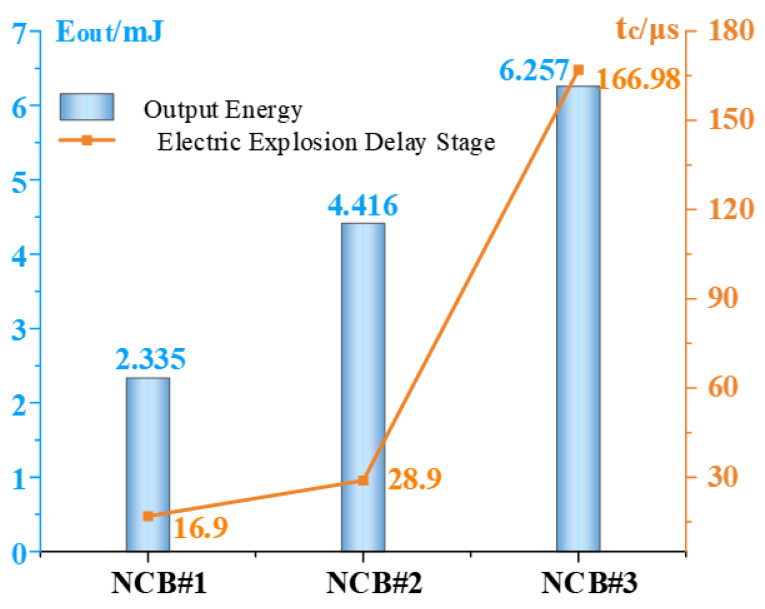
Electrical explosion delay times and output energies of the NCBs.

**Figure 9 micromachines-14-00961-f009:**
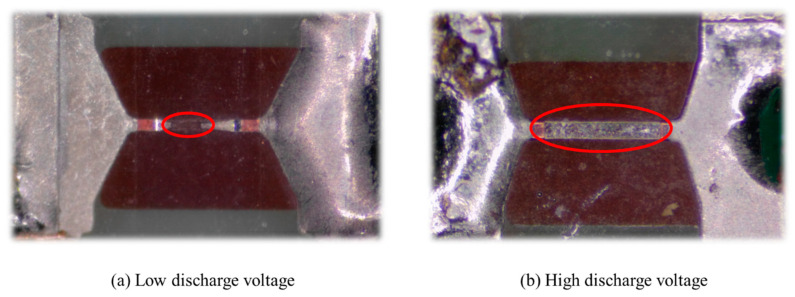
Bridge morphology under the action of different charging voltages. (**a**) Bridge area morphology after low voltage discharge, (**b**) Bridge area morphology after high voltage discharge.

**Figure 10 micromachines-14-00961-f010:**
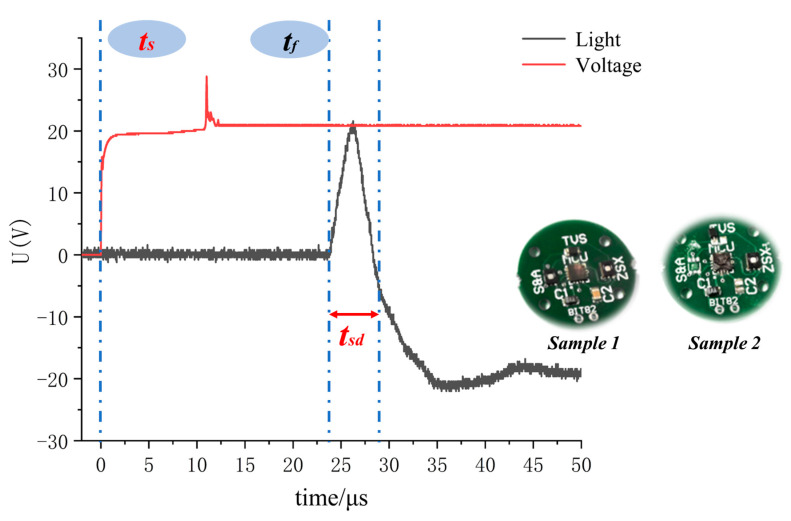
Response time and damage effect test results for the self-destruction device.

**Table 1 micromachines-14-00961-t001:** The action conditions and time scale of different self-destruction devices.

Type of Self-Destruct Chip	Driven Conditions	Action Time
Biodegradable CMOS chip [9]	Not explained	Within 2 h
Explosive nano-energetic film [13]	Voltage: 20 V	Within 1 s
Nano-thermites [14,15,16,17]	Not explained	From milliseconds to tens of seconds
Explosive materials [18,19,20]	Capacitor discharge: 15 V 33 μF	Microsecond

## Data Availability

All other data are available from the corresponding authors upon reasonable request.

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
