# Peer review of "Research on Energetic Micro-Self-Destruction Devices with Fast Responses"

_micromachines, 2023, doi:10.3390/mi14050961_

Round 1

Reviewer 1 Report

In this study, the self-destruction device was proposed using nichrome (Ni-Cr) bridge. This work seems novel and important, so I recommend this paper to be published after minor revision.

1. How do characteristics change as devices are scaled down for higher chip density? (in terms of operation voltage/current, energy consumption, and delay)

2. Could you compare the energy consumption and delay with other self-destruction devices?

Author Response

Question1:  How do characteristics change as devices are scaled down for higher chip density? (in terms of operation voltage/current, energy consumption, and delay)

Answer1: Thanks for your comments. But We have not yet done research on this issue, we will consider your question in the future work.

Question2:  Could you compare the energy consumption and delay with other self-destruction devices?

Answer2: In view of your opinion, the comparison table of driving condition and action time of different self-destruction devices has been supplemented on the second page of this paper. See Table 2 for details.

Reviewer 2 Report

The paper By Kan and co-workers reports experiments dedicated to the self-destruction of devices as ultimate security modules to protect information data. The paper is well written and well-structured for the readership in the community. The state of the art and question raised are very clearly presented. I particularly appreciate the effort to provide modelling in addition to experiment. For general strengthening of the paper conclusions, and didactive effort so that the paper is useful to the community, I would suggest the following major revisions, which all are more or less oriented to better description and quantification of the results.

1.       The paper introduction is more or less a state of the art. The use of energetic materials for self-destruction is not new and some effort in the US and recently in Europe have been made using alternative non-detonating thermite materials, some being hybridized with gas generating materials. Their interest is the technological compatibility with MEMs technologies, also the materials are safe and non-toxic. I would suggest the authors to mention them to offer a complete view of the existing approaches, which is feasible in this application.

S. S. Pandey, C. H. Mastrangelo, An Exothermal Energy Release Layer for Microchip
Transience, IEEE Sens. J. (2013), 1759-1762, https://doi.org/10.1109/ICSENS.2013.6688572.

Developing a highly responsive miniaturized security device based on a printed copper ammine energetic composite F Sevely, T Wu, FSF de Sousa, L Seguier, V Brossa, S Charlot, A Esteve, ...

Sensors and Actuators A: Physical 346, 11383 (2022)

. J. Fleck, R. Ramachandran, A. K. Murray, W. A. Novotny, G. T. C. Chiu, I. E. Gunduz, S. F.
Son, J. F. Rhoads, Controlled Substrate Destruction Using Nanothermite, Propellants Explos.
Pyrotech. 42 (6) (2017), 579-584, https://doi.org/10.1002/prep.201700008

S. S. Pandey, N. Banerjee, Y. Xie, C. H. Mastrangelo, Self-Destructing Secured Microchips by
On-Chip Triggered Energetic and Corrosive Attacks for Transient Electronics, Adv. Mater. Technol.
3 (7) (2018), 1800044, https://doi.org/10.1002/admt.201800044

2.       The way the simulation is performed and its description is rather crude. Please describe in a more pedagogic way the input and output features. How A, B, C and all dimensionless parameters are chosen? Add the gap size in the schematics in Fig. 4. The unit in Fig. 5 is not clear and should be given in standard one. Same goes for Fig. 6.

3.       Altogether, modelling plus experiment offer a good overview of aspects of the system under consideration. However, is there a way to connect quantitatively the modelling and experiment? This should be at least commented and/or suggested for future work.

4.       The way the ship is destroyed or damage is not discussed with accuracy. Quantification of the destruction should be provided on a test study: photos, electrical tests …

Author Response

  1. Literature has been added to the article, [14]-[17].
  2. In formula ( 1 ), p is the detonation pressure ; V is the relative specific volume of detonation products ; E is the internal energy per unit volume ; A, B, R1, R2 and ω are material constants.

The picture has been changed.

  1. Your problem is the next research content of our research group, which combines simulation and experiment to get a more accurate design method.
  2. The output voltage of the chip is collected by the oscilloscope. The normal chip works as shown in Figure 1, and the electrical signal after the chip self-destruction is shown in Figure 2. After the chip self-destructs, the set signal cannot be output.

Reviewer 3 Report

The paper introduced a novel Self-Destruction devices with fast responses. The topic is of vital importance for the furture safety of information. The work is well designed and disscussed. It think it can be accpet in the current form.

Author Response

Thank you for your comments.

Reviewer 4 Report

Summary:

The authors present a device that is capable of generating GPa level detonation waves through the explosion of energetic materials to cause severe damage to the storage chips. The authors have done a decent study but will need address some issues before acceptance.

Minor Comments:

1.       Figures appear pixelated when zoomed in. Best to use vector graphics to make nice figures. But this is a small issue.

2.       Some of the parameters in table within Figure 4 don’t have symbols. Tables need to remade more cleanly.

Major Comments:

1.       Many of the devices in defense are exposed to extreme environments. Are there any physical conditions (extreme shock/temperature, pressure etc.) that can set off the device unintentionally?

2.       The author state various material models for air and state equation etc. in all caps. Are there references for these models? If so, can they please cite it appropriately?

3.       I would rather they delineate tables/figures in Figure 4.

4.       Terms in equation 1 are not explained correctly in the text, please elaborate what these terms are. What is R1, R2 and w?

5.       Why is Co,c1,c2,c3 and c6 0? If it is zero, what is the point of equation 2? Should they simply it to this present case?

6.       The authors loosely use pressure and stress at several places (including within figures). Please be more precise.

7.       Are there any residual stresses in the device during or after fabrication? 

It's okay. 

Author Response

Minor Comments:

Question1:  Figures appear pixelated when zoomed in. Best to use vector graphics to make nice figures. But this is a small issue.

Answer1:According to the comment, this paper had modified the images (Figure 4. Simulation model and related parameters. Figure 6. Detonation wave pressure values under different gap conditions and the associated transfer process.).

Question2:  Some of the parameters in table within Figure 4 don’t have symbols. Tables need to remade more cleanly.

Answer2: Equation(2) is used to describe the air model in the detonation field, C0–C6 are the coefficients of the state equation of air, these parameters are dimensionless.

Major Comments:

Question1: Many of the devices in defense are exposed to extreme environments. Are there any physical conditions (extreme shock/temperature, pressure etc.) that can set off the device unintentionally?

Answer: In practical applications, there are unexpected stimuli that make the device work. Electrostatic is the major stimuli to ignited energtic materials, the security is researched by references [14] and [16] , which have been cited in this paper.

Question2: The author state various material models for air and state equation etc. in all caps. Are there references for these models? If so, can they please cite it appropriately?

Answer: The references for various material models has been added in paper, reference[25] and reference [26].

Question3: I would rather they delineate tables/figures in Figure 4.

Answer: The parameter equations and meanings of each material have been supplemented on page 5.

Question4 to Question5: Terms in equation 1 are not explained correctly in the text, please elaborate what these terms are. What is R1, R2 and w? Why is Co,c1,c2,c3 and c6 0? If it is zero, what is the point of equation 2? Should they simply it to this present case?

Answer: The parameters (R1, R2 and w) have been described in manuscript, on page 5. “In equation (1), p is the detonation pressure; V is the relative specific volume of detonation products; E is the internal energy per unit volume; A, B, R1, R2 and ω are material constants.” Relevant literature has been supplemented in the article, [25] and [26].

Question 6: The authors loosely use pressure and stress at several places (including within figures). Please be more precise.

Answer: The stress value has been changed to the pressure value.

Question 7: Are there any residual stresses in the device during or after fabrication?

Answer: There is no residual stress in the process of making the equipment.

Round 2

Reviewer 4 Report

All comments addressed.